# DNA Methylation of Imprinted Genes *KCNQ1*, *KCNQ1OT1*, and *PHLDA2* in Peripheral Blood Is Associated with the Risk of Breast Cancer

**DOI:** 10.3390/cancers14112652

**Published:** 2022-05-27

**Authors:** Jinming Fu, Lei Zhang, Dapeng Li, Tian Tian, Xuan Wang, Hongru Sun, Anqi Ge, Yupeng Liu, Xianyu Zhang, Hao Huang, Shuhan Meng, Ding Zhang, Liyuan Zhao, Simin Sun, Ting Zheng, Chenyang Jia, Yashuang Zhao, Da Pang

**Affiliations:** 1Department of Epidemiology, College of Public Health, Harbin Medical University, Harbin 150081, China; fu_jinming@hrbmu.edu.cn (J.F.); zhanglie@hrbmu.edu.cn (L.Z.); ldroc1987@gmail.com (D.L.); 102415@hrbmu.edu.cn (T.T.); 201701033@hrbmu.edu.cn (X.W.); sunhongru_1024@126.com (H.S.); anqi_ge@hrbmu.edu.cn (A.G.); liuyupeng@wmu.edu.cn (Y.L.); huanghao@hrbmu.edu.cn (H.H.); mengsh@hrbmu.edu.cn (S.M.); zhangd_95@hrbmu.edu.cn (D.Z.); 2018020181@hrbmu.edu.cn (L.Z.); sunsimin@hrbmu.edu.cn (S.S.); zitty@hrbmu.edu.cn (T.Z.); jiachenyang1126@hrbmu.edu.cn (C.J.); 2Department of Biostatistics, School of Public Health, Xuzhou Medical University, Xuzhou 221004, China; 3Department of Breast Surgery, Harbin Medical University Cancer Hospital, Harbin Medical University, Harbin 150081, China; zhangxianyu@ems.hrbmu.edu.cn

**Keywords:** breast cancer, DNA methylation, peripheral blood leukocytes, imprinted gene, *KCNQ1OT1*

## Abstract

**Simple Summary:**

DNA methylation alterations of imprinted genes lead to loss of imprinting, and studies have explored the mechanism of the loss of imprinting in breast cancer development. However, the association between the methylation of imprinted genes in peripheral blood and the risk of breast cancer is largely unknown. Our study is the first to identify the CpG sites of imprinted genes associated with the risk of breast cancer by utilizing HumanMethylation450 data from public datasets. We discovered and validated that peripheral blood DNA methylation of *KCNQ1*, *KCNQ1OT1*, and *PHLDA2* at chromosome 11p15.4-15.5 is associated with breast cancer susceptibility. Individuals with *KCNQ1OT1*-region hypermethylation (>0.474) had a lower risk of breast cancer. Additionally, the methylation level of *KCNQ1OT1* was also unaffected by leukocyte heterogeneity. In summary, the *KCNQ1OT1* region could be a potential biomarker for breast cancer risk assessment.

**Abstract:**

Methylation alterations of imprinted genes lead to loss of imprinting (LOI). Although studies have explored the mechanism of LOI in breast cancer (BC) development, the association between imprinted gene methylation in peripheral blood and BC risk is largely unknown. We utilized HumanMethylation450 data from TCGA and GEO (*n* = 1461) to identify the CpG sites of imprinted genes associated with BC risk. Furthermore, we conducted an independent case-control study (*n* = 1048) to validate DNA methylation of these CpG sites in peripheral blood and BC susceptibility. cg26709929, cg08446215, cg25306939, and cg16057921, which are located at *KCNQ1*, *KCNQ1OT1*, and *PHLDA2*, were discovered to be associated with BC risk. Subsequently, the association between cg26709929, cg26057921, and cg25306939 methylation and BC risk was validated in our inhouse dataset. All 22 CpG sites in the *KCNQ1OT1* region were associated with BC risk. Individuals with a hypermethylated *KCNQ1OT1* region (>0.474) had a lower BC risk (OR: 0.553, 95% CI: 0.397−0.769). Additionally, the methylation of the *KCNQ1OT1* region was not significantly different among B cells, monocytes, and T cells, which was also observed at CpG sites in *PHLDA2*. In summary, the methylation of *KCNQ1*, *KCNQ1OT1*, and *PHLDA2* was associated with BC risk, and *KCNQ1OT1* methylation could be a potential biomarker for BC risk assessment.

## 1. Introduction

Breast cancer (BC) is one of the most common malignancies, with the highest incidence in women [1]. According to GLOBOCAN 2020, there were approximately 2.3 million newly diagnosed cases of female BC globally in 2020 [1]. In China, the incidence of female BC is lower than in Western countries; however, it has presented a gradual upward trend in the last decade [2]. Genetic variation, such as *BRCA1* and *BRCA2* mutations, has been closely linked to the development of BC but only accounts for 5–10% of BC cases [3,4,5]. In contrast, epigenetic changes, including DNA methylation alterations, are more frequent, and they usually occur in the early stages of BC, which gives them the potential to become biomarkers for BC risk prediction [6].

Genomic imprinting is an epigenetic marker that is established within germ cells through DNA methylation [7]. At the imprinted locus, genes are expressed monoallelically based on their parental origin [8]. Imprinted genes are usually distributed in clusters, and their expression is regulated by imprinted control regions that are essentially differentially methylated regions (DMRs) [9]. Loss of imprinting (LOI), as an early-occurring aberration in cancer [10], is usually accompanied by methylation level changes, thereby leading to biallelic expression or completely silencing [11]. As imprinted genes are expressed monoallelically, disruption of imprinting is more likely to increase the tumor susceptibility of individuals [12]. Accumulating studies have found that aberrant DNA methylation of imprinted genes may accelerate the process of BC. A nested case-control study of 40 primary invasive BC patients identified that *KCNQ1* could be used to distinguish patients who developed and did not develop contralateral BC [13]. *KCNQ1OT1* is an unspliced long noncoding RNA that plays a role in regulating the cell cycle, migration, and invasion, inhibiting cell apoptosis and thus promoting BC growth [14]. Hypomethylation of the *KCNQ1OT1* promoter results in its high expression, causing downregulation of *CDKN1C* expression, thereby accelerating the development of BC [15]. *PHLDA2* is an important suppressor gene whose expression is regulated by *KCNQ1OT1* [16]. Moon et al. found that the expression of *PHLDA2* can be used as a novel prognostic biomarker of triple-negative BC [17]. In addition, regulation of *IGF2* and *ARHI* expression by DNA methylation was also observed in BC tissues and cell lines, as well as differences in methylation of *PEG3*, *MEST*, and *GNAS* between cancer and normal tissues [18,19,20,21,22,23]. All these findings imply that the methylation of imprinted genes could be a potential biomarker for assessing BC risk. In the early detection of BC, tissue specimens are difficult to obtain. In contrast, the detection of peripheral blood DNA methylation is more feasible for predicting the risk of BC, given the advantages of less invasiveness and easy operation [24,25]. At present, more than 100 human imprinted genes have been identified; however, there are only a few studies on the relationship between the methylation of imprinted genes in peripheral blood leukocytes and the risk of BC [26]. There is an urgent need to systematically explore new methylation-based imprinted gene markers in peripheral blood for BC risk assessment.

Therefore, this study aimed to systematically discover imprinted genes associated with BC risk using high-throughput public data and validate whether these genes are promising biomarkers for BC risk assessment in an independent case-control study.

## 2. Materials and Methods

### 2.1. Public Data Resource

Public datasets were downloaded from The Cancer Genome Atlas (TCGA, https://xena.ucsc.edu/, accessed on 24 December 2021) and the Gene Expression Omnibus (GEO, https://www.ncbi.nlm.nih.gov/geo/, accessed on 30 December 2021) (Appendix A). The GSE74738 dataset includes five diandric (two paternal copies of the genome and one maternal) and five digynic (two maternal copies of the genome and one paternal) triploids and was used to discover differentially methylated CpG sites of imprinted genes in human embryos. The TCGA BC methylation dataset (*n* = 878) of the Infinium 450K array and the gene expression profile were downloaded to identify the CpG sites of imprinted genes associated with BC. The GSE51032 dataset (*n* = 573), which was derived from a nested case-control study, contains peripheral blood DNA methylation data from BC patients and healthy controls, and it was used to identify the CpG sites of imprinted genes associated with BC risk. The methylation level of CpG sites was represented by a β value, which was calculated as the methylated probe intensity (M) divided by the overall intensity (the sum of the methylated and unmethylated probe intensities).

### 2.2. Selection of CpG Sites of Imprinted Genes Associated with BC Risk

The workflow is presented in Figure 1. Differential methylation analyses between diandric and digynic triploids were first performed by using the Bioconductor R package “ChAMP” (Figure 1a). A differentially methylated position was defined as a CpG with the absolute value of methylation difference (|Δβ|) ≥ 0.05 and adjusted *p*-value < 0.15. A DMR was defined as a region containing at least three differentially methylated CpG sites within 300 bp that had an adjusted *p*-value < 0.05 and |Δβ| ≥ 0.05. Considering that the imprinted control regions, which are essentially DMRs, play an important role in the regulation of imprinted gene expression, the differentially methylated positions encompassed in DMRs were selected as differentially methylated CpG sites in embryos. Then, these CpG sites were integrated with an online imprinted gene database (geneimprint, https://www.geneimprint.com, accessed on 12 January 2022) to identify differentially methylated CpG sites of imprinted genes in human embryos (Figure 1a).

Student’s *t*-test was used to evaluate the methylation differences in selected CpG sites between 782 BC tissues and 96 normal tissues in the TCGA dataset and between 233 BC patients and 340 healthy controls in the GSE51032 dataset (Figure 1b,c). Those CpG sites with significant methylation differences and the same differential direction both in tissue and peripheral blood (*p*-value < 0.05 and |Δβ| ≥ 0.01) were the potential CpG sites of imprinted genes associated with BC risk.

To explore the potential regulatory function of selected CpG sites, the Pearson correlation method was also applied to assess the correlation between the methylation of selected CpG sites and corresponding expression in the TCGA dataset.

### 2.3. Inhouse Validation Study

To further validate the association between the methylation of selected CpG sites and BC risk, we performed an independent case-control study utilizing quantitative methylation-level detection (Figure 1d). In total, 523 primary BC patients and 538 controls were recruited for the case-control study. All cases were newly diagnosed female BC patients (including invasive ductal carcinoma and ductal carcinoma in situ) who were hospitalized in the Tumor Hospital of Harbin Medical University between 2010 and 2014. Patients with metastatic BC were excluded from the study. All the included controls were female patients who were hospitalized in the orthopedic and ophthalmology department of the Second Affiliated Hospital of Harbin Medical University during the same period and female volunteers from the Xiangfang district of Harbin. All subjects’ demographic information and family history of cancer were obtained through interviews with trained investigators. Additionally, all the controls were asked for their disease history to exclude individuals with a history of any cancers.

### 2.4. DNA Extraction and Bisulfite Conversion from Peripheral Blood

DNA was extracted from peripheral blood leukocytes using a QIAamp DNA Blood Mini Kit (Qiagen, Hilden, Germany), and the concentration of extracted DNA was measured using a Nanodrop 2000 Spectrophotometer (Thermo Scientific, Waltham, MA, USA). Genomic DNA was bisulfite-converted using an EpiTect Bisulfite kit (Qiagen, Hilden, Germany). Bisulfite-converted DNA was stored at −20 °C for targeted bisulfite sequencing. All experimental procedures were performed in accordance with the manufacturer’s protocols.

### 2.5. Cell Fractionations from Peripheral Blood Leukocytes

Peripheral blood specimens were collected from 10 newly diagnosed female BC patients from the Tumor Hospital of Harbin Medical University and from 10 healthy female volunteers from Harbin Medical University. Leukocytes were freshly isolated from peripheral blood using a Human Peripheral Blood Leukocyte Isolation Kit (Solarbio, Beijing, China) within two hours after blood collection. Isolated leukocytes were suspended in 1 mL PBS for the following cell fractionations. B cells were isolated from leukocytes first using Dynabeads^®^ CD19 pan B (Invitrogen, Carlsbad, CA, USA). Then, monocytes were isolated from B cell-depleted leukocytes using Dynabeads^®^ CD14 (Invitrogen, Carlsbad, CA, USA). Finally, T cells were purified from B-cell/monocyte-depleted leukocytes using Dynabeads^®^ CD3. DNA extraction and bisulfite conversion from B cells, T cells, and monocytes were performed immediately after purification and the experimental procedures have been described above.

### 2.6. Quantitative Methylation Analysis

The DNA methylation levels of selected CpG sites in the peripheral blood leukocyte specimens were detected using MethylTarget™ (Genesky Biotechnologies Inc., Shanghai, China). As a previous study has described, this is a next-generation sequencing technology for methylation profiling of specific CpG sites/regions [27]. The primers were designed using Primer3 software (Appendix A) [28]. A two-step PCR was performed for each bisulfite-converted DNA sample, including a first step for amplifying the targeted DNA sequence and a second step for adding barcodes. High-throughput sequencing was performed on the Illumina Hiseq platform using a 2 × 150 bp paired-end sequencing mode. The detailed procedures are described in the Appendix A.

### 2.7. Statistical Analysis

All genome coordinates were based on the human genome version GRCh37/hg19. The R packages “mice” and “VIM” were used to analyze the missing patterns of demographic information and to fill the missing data with multiple imputations for subsequent analyses [29]. The differences in demographic information between BC patients and controls were evaluated using Student’s *t*-tests for normally distributed continuous variables, *χ^2^* tests for categorical variables, and the Mann–Whitney U tests for non-normally distributed continuous variables.

Considering that targeted bisulfite sequencing could detect the methylation levels of selected CpG sites and their surrounding 150 bp, differential methylation analyses were performed first for each CpG site involved in the region of the selected imprinted genes. Student’s *t*-test was used to evaluate the significant differences in the methylation levels of each CpG site between BC patients and controls in the inhouse validation. The optimal cutoff values of the methylation level were calculated using the receiver operating characteristic curve, and they were applied to categorize all participants into hypomethylated and hypermethylated groups. Univariate and multivariate logistic regression analyses were conducted to analyze the association between the peripheral blood methylation of each CpG site and the risk of BC. The results are represented as odds ratios (ORs) and 95% confidence intervals (95% CIs).

The Spearman correlation method was applied to assess the co-methylation of adjacent CpG sites, and the co-methylation patterns were visualized by the R package “coMET” [30]. If the methylation levels of multiple adjacent CpG sites in the same region were highly correlated, the weighted methylation level of this region was calculated using the following formula [31]:(1)$$\mathrm{weighted}\;\mathrm{methylation}\;\mathrm{level}=\overset{n}{\underset{i=1}{\sum }}\mathrm{Ci}/\overset{n}{\underset{i=1}{\sum }}\left(\mathrm{Ci}+\mathrm{Ti}\right)$$

Here,$\;\overset{n}{\underset{i=1}{\sum }}\mathrm{Ci}$ represents the sum of the methylated reads of all CpG sites involved in this region, and $\overset{n}{\underset{i=1}{\sum }}\left(\mathrm{Ci}+\mathrm{Ti}\right)$ represents the sum of the overall reads of all CpG sites involved in this region. Then, univariate and multivariate logistic regression was performed to analyze the association between the weighted methylation levels of imprinted gene regions and the risk of BC.

Subgroup analyses were also conducted to assess the BC susceptibility of significant CpG sites in different subgroups defined by age and molecular type [32]. In addition, the methylation levels of BC-associated CpG sites and regions among T cells, B cells, and monocytes were compared to evaluate the methylation heterogeneity of these CpG sites and regions in different leukocyte subtypes.

All statistical analyses were conducted with R version 4.0.3 software (Institute for Statistics and Mathematics, Vienna, Austria). Two-sided *p* < 0.05 was considered statistically significant.

## 3. Results

### 3.1. Discovery of the CpG Sites of Imprinted Genes Associated with BC Risk

In the GSE74738 dataset, 31,269 differentially methylated CpG sites were found between diandric and digynic triploids, including 12,482 paternal hypomethylated CpG sites and 18,787 maternal hypomethylated CpG sites (Figure 2a). These differential CpG sites were distributed on 22 chromosomes and corresponded to 8837 genes (Figure 2b). Then, 3596 CpG sites encompassed in the DMRs were integrated with known imprinted genes, and 449 CpG sites (corresponding to 32 imprinted genes) were identified as differentially methylated CpG sites of imprinted genes in human embryos (Figure 2c,d).

In the TCGA dataset, the methylation levels of 299 CpG sites were significantly different between BC and normal tissues, including 149 hypermethylated CpG sites and 150 hypomethylated CpG sites (Figure 2e). Among the 26 CpG sites differentially methylated between BC patients and healthy controls in the GSE51032 dataset, 16 CpG sites presented the same differential direction as in the tissues (Appendix A, Figure 2e). These 16 CpG sites (corresponding to 12 imprinted genes) were identified as differentially methylated CpG sites of imprinted genes associated with BC risk (Figure 2f).

Half of the above CpG sites were annotated on five imprinted genes (*CDKN1C*, *H19*, *KCNQ1*, *KCNQ1OT1*, and *PHLDA2*), which were located on chromosome 11p15.4-15.5. Chromosome 11p15.4-15.5 contains a large number of imprinted genes, and their aberrant methylation may contribute to the development of BC [15,33,34]. Many studies have been conducted on the different mechanisms of action of *H19* in human BC [35], while only a few studies at the cell-line and tissue levels have been carried out to explore the roles of *CDKN1C*, *KCNQ1*, and *KCNQ1OT1* in BC. Therefore, *CDKN1C* (cg04402633 and cg11744767), *KCNQ1* (cg26709929), *KCNQ1OT1* (cg08446215 and cg25306939), and *PHLDA2* (cg16057921) were retained for subsequent validation.

cg08446215, cg25306939, cg11744767, and cg16057921 were located in CpG islands, cg04402633 was located on the North Shore, and cg26709929 was located in the Open Sea. In the nested case-control study from GSE51032, increased methylation levels of cg11744767, cg16057921, and cg04402633 were associated with a higher risk of BC, while the increased methylation levels of cg26709929, cg08446215, and cg25306939 may have reduced the risk of BC (Appendix A). In 854 paired methylation and expression data entries from TCGA, the methylation levels of cg11744767, cg08446215, cg25306939, and cg16057921 at CpG islands were negatively correlated with their gene expression, and the methylation level of cg26709929 positively correlated with its gene expression (Appendix A).

### 3.2. The Methylation of Individual CpG Sites of Selected Imprinted Genes and BC Risk in the Validation Set

Due to the high density of CGs, the primer design for the target region covering cg11744767 failed. Additionally, six samples with failed PCR amplification and seven samples with a bisulfite conversion rate below 95% were filtered out. Targeted bisulfite sequencing successfully detected the methylation levels of five CpG sites (cg04402633, cg26709929, cg08446215, cg25306939, and cg16057921) and their surrounding 150 bp in 1048 samples from 514 BC patients and 534 controls in the validation set (Appendix A).

The mean ages of BC patients and controls were 52.0 ± 9.35 and 52.1 ± 10.76, respectively. Compared with controls, a higher proportion of BC patients had a family history of BC and other cancers. The detailed demographic characteristics of BC patients and controls are presented in Table 1.

For targeted bisulfite sequencing, the methylation levels of CpG sites were relatively stable when the depth was over 1000X. Therefore, quality control of samples (depth ≥ 1000X) was performed based on different target regions. Finally, data from 1019 samples for the region covering cg26709929, data from 1043 samples for the region covering cg08446215, data from 663 samples for the region covering cg25306939, and data from 1030 samples for the region covering cg16057921 were retained for the subsequent analyses (Appendix A). The demographic characteristics of BC patients and controls for each target imprinted-gene region are presented in Appendix A.

In the target region of *KCNQ1*, the methylation level of cg26709929 was slightly lower in BC patients than in controls, and a similar result was observed in the CpG with genome coordinates 2482233. After adjusting for age, BMI, ethnicity, location, family history of BC, and family history of other cancers, the ORs for increased methylation levels per SD were 0.852 (95% CI: 0.723–0.985, *p* = 0.047) and 0.870 (95% CI: 0.765–0.989, *p* = 0.033), respectively (Appendix A).

The target region 1 of *KCNQ1OT1* covered 10 CpG sites. The methylation levels of all CpG sites showed no significant differences between BC patients and controls, and the results of multivariate logistic regression indicated that no CpG sites were associated with the risk of BC (Appendix A). In target region 2 of *KCNQ1OT1*, the methylation level of cg25306939 was lower in BC patients than in controls (Δβ = −0.011). Similar results were also observed in 21 adjacent CpG sites (Δβ ranged from −0.031 to −0.011). In addition, multivariate logistic regression analyses revealed that the methylation levels of all 22 CpG sites were associated with the risk of BC; that is, a higher methylation level in a CpG sites was associated with a lower risk of BC (Table 2).

There were also two BC risk-associated CpG sites in the target region of *PHLDA2*, with small differences in methylation between BC patients and controls. With the increasing methylation levels of the two CpG sites in the target region of *PHLDA2*, the risk of BC also increased gradually. The ORs for increased methylation levels per SD were 1.140 (95% CI: 1.010–1.300, *p* = 0.041) and 1.150 (95% CI: 1.010–1.300, *p* = 0.036) (Appendix A).

In summary, the association between the methylation of cg08446215, cg25306939, and cg16057921 and BC risk was validated. When using the optimal cutoff to separate BC patients and controls into hypomethylated and hypermethylated groups (Appendix A), differentially methylated CpG sites in the target regions of *KCNQ1OT1* and *PHLDA2* were still associated with the risk of BC (Appendix A).

### 3.3. Methylation of the KCNQ1OT1 Target Region and BC Risk in the Validation Set

The methylation levels of 22 CpG sites in *KCNQ1OT1* region 2 presented strong correlations, which indicated that these CpG sites were in a co-methylation pattern (Figure 3a). Therefore, the weighted methylation level was calculated to investigate the association between the methylation of the *KCNQ1OT1* region and the risk of BC. In the multivariate logistic analysis, the OR for a weighted methylation-level increase per SD was 0.819 (95% CI: 0.697–0.960, *p* = 0.014). Individuals were separated into a hypomethylated group (<0.474) and hypermethylated group (≥0.474) based on the optimal cutoff value for weighted methylation of the *KCNQ1OT1* region (Appendix A). Compared with the hypomethylated group, individuals in the hypermethylated group had a 44.7% lower risk of BC (Figure 3b).

### 3.4. Subgroup Analysis

In the subgroup analysis defined by age, individuals were separated into a younger group (<50 years old) and an older group (≥50 years old). The methylation of cg26709929 in *KCNQ1* was not associated with BC risk in either subgroup, which was also observed for another CpG with genome coordinates 2,482,233 (Appendix A). In the younger subgroup, individuals with hypermethylation of *KCNQ1OT1* region 2 had a lower risk of BC (Table 3). In the older subgroup, individuals with hypermethylation of two CpG sites in the target region of *PHLDA2* had a higher risk of BC (Appendix A).

According to the expression of estrogen receptor, progesterone receptor, and HER-2, BC was classified into four subtypes, namely luminal A, luminal B, HER-2 overexpressing, and Basal-like. The methylation of cg26709929 on *KCNQ1* was associated with the risk of the HER-2-overexpressing subtype of BC, while the methylation of CpG with genome coordinates 2,482,233 was associated with the risk of the luminal B subtype of BC (Appendix A). The methylation of two CpG sites in the target region of *PHLDA2* was also associated with the risk of the luminal B subtype of BC (Appendix A). Moreover, the weighted methylation level of *KCNQ1OT1* region 2 was associated with the risk of both the luminal A and HER-2-overexpressing subtypes of BC, with adjusted ORs of 0.446 (95% CI: 0.276–0.706, *p* < 0.001) and 0.424 (95% CI: 0.176–0.936, *p* = 0.042), respectively (Table 3).

### 3.5. The Methylation Levels of KCNQ1, KCNQ1OT1, and PHLDA2 in Different Leukocyte Subtypes

The weighted methylation level of *KCNQ1OT1* region 2 was not significantly different among B cells, monocytes, and T cells (Figure 4a,b). For BC-associated CpG sites in the target region of *PHLDA2*, similar methylation levels were also observed in B cells, monocytes, and T cells (Figure 4c–f). Although the methylation levels of BC-associated CpG sites in the target region of *KCNQ1* were found to be significantly different between T cells and monocytes and between B cells and monocytes, the difference may not affect the overall methylation level of peripheral blood due to the small proportion of monocytes (Appendix A).

## 4. Discussion

By systematically integrating HumanMethylation450 data from TCGA and GEO, we screened six CpG sites of imprinted genes potentially associated with BC risk. We further validated the association between the methylation of *KCNQ1*, *KCNQ1OT1*, and *PHLDA2* in peripheral blood leukocytes and the risk of BC using independent external data. Our study indicated that decreased methylation of *KCNQ1* and *KCNQ1OT1* and increased methylation of *PHLDA2* were associated with an increased risk of BC. Individuals with hypomethylation of *KCNQ1OT1* (<0.474) have a 1.808 times higher risk of BC than those with hypermethylation. The weighted methylation level of the *KCNQ1OT1* region was also associated with the risk of luminal A and HER-2-overexpressing subtypes of BC. In addition, the methylation level of the *KCNQ1OT1* region showed no significant differences in T cells, B cells, or monocytes. These results highlight that the methylation of imprinted genes in the 11p15.4-15.5 region, especially *KCNQ1OT1* methylation, has strong potential as a biomarker for BC risk assessment.

DNA methylation is the main epigenetic regulation mechanism of genomic imprinting. The methylation of imprinted genes is established in gametes relying on DNMT3a and DNMT3b, and it is maintained by DNMT1 in somatic cells [36]. DNA methylation can be transmitted from cell to cell, but it is not permanent. Some changes in DNA methylation can result in various human disorders, including cancer [37]. LOI occurs frequently and precociously in tumors, making it a potential candidate marker for assessing the risk of multiple cancers [10]. LOI includes gain or loss of methylation, which may cause activation of the normally silenced allele or silencing of the normally active allele [38]. Although numerous studies have explored the mechanism of LOI in BC development, the association between LOI and the risk of BC remains to be demonstrated in population studies.

The soma-wide nature of imprinting means that the DNA methylation alterations of imprinted genes are not limited to BC tissues, thus allowing multiple tissues (including peripheral blood) to share “early markers of carcinogenesis” [26,39]. The detection of biomarkers in peripheral blood is minimally invasive and highly compliant. Public genome-wide methylation data from TCGA and GEO also provide the opportunity to discover new BC-associated biomarkers. In our study, by integrating the largest methylation datasets, *KCNQ1*, *KCNQ1OT1*, and *PHLDA2* located at chromosome 11p15.4-15.5 were identified to be associated with BC risk. The methylation levels of *KCNQ1* and *KCNQ1OT1* were lower in BC patients than in controls, which also validated the mechanism reported in a previous study; that is, loss of methylation at chromosome 11p15.5 is common in human adult tumors [34].

High-throughput methylation detection technologies, such as 450K and 850K arrays, are usually used for marker screening due to their complex technology and high price. In contrast, targeted bisulfite sequencing is more suitable to verify candidate markers and apply them in clinical practice, and it has the advantage of being highly specific for targeted regions at a single-base level, requiring a low amount of genomic DNA and multigene profiling. By using targeted bisulfite sequencing, our study validated the hypomethylation status of the *KCNQ1OT1* region in BC patients and its association with BC risk. Hypomethylation of the *KCNQ1OT1* region (located in the promoter) may result in its high expression, thereby contributing to BC development. Based on these results, our study suggested that *KCNQ1OT1* methylation could be a potential new biomarker in the clinical field for predicting BC risk. The slight differences in the methylation levels of *KCNQ1* and *PHLDA2* between BC patients and controls also implied their limited clinical utility.

BC is a highly heterogeneous disease. The expression of hormone receptors (such as estrogen receptor and progesterone receptor) is implicated in the progression of BC [40]. The subgroup analyses in this study indicated that *KCNQ1OT1* methylation has the potential to help in assessing the risk of luminal A and HER-2-overexpressing subtypes of BC, while *PHLDA2* methylation could also be a novel marker to assess the risk of luminal B subtype BC.

Similar to tumor tissues, peripheral blood leukocytes consist of different cell types that may carry different methylation levels and whose relative proportions may vary between individuals [41]. A cross-sectional analysis indicated that no significant change in the methylation levels of imprinted genes was observed between 25 and 87 years of age [26]. In the BC-free individuals from the validation dataset, we observed that the methylation levels of *KCNQ1*, *KCNQ1OT1*, and *PHLDA2* did not change significantly with ages ranging from 25 to 78 years (Appendix A). The results of methylation sequencing in different leukocyte subtypes also presented no significant differences in methylation levels of *KCNQ1OT1* and *PHLDA2* among B cells, monocytes, and T cells, which validated the point that the methylation of imprinted genes is unaffected by cellular heterogeneity [42]. All these results support the idea that the methylation of imprinted genes could be a potential biomarker for assessing the risk of BC.

Compared with previous studies, this study has several advantages. Our study, for the first time, systematically screened BC-associated imprinted genes based on high-throughput array data. The associations between the methylation of selected imprinted genes in peripheral blood and the risk of BC were further validated in an independent case-control study using targeted bisulfite sequencing. The methylation levels in T cells, B cells, and monocytes were also compared to evaluate the methylation stability of selected imprinted genes in leukocytes.

Our study also has some limitations. First, the validation set was a case-control study, which means that the causality between imprinted gene methylation and the risk of BC could not be determined. However, we believe that the methylation of *KCNQ1*, *KCNQ1OT1*, and *PHLDA2* is worthy of further evaluation in prospective studies, as the BC-associated CpG sites of imprinted genes were screened based on the GSE51032 set, which was designed as a nested case-control study. Second, all the BC patients for the validation set came from a single center, which may have introduced selection bias. Third, the methylation differences in the target CpG sites in this study were relatively small between BC patients and controls. This result suggests that we need to be cautious in drawing conclusions from our study, and it is necessary to validate the differences in methylation levels for these CpG sites in peripheral blood and their association with BC risk in other studies.

## 5. Conclusions

Our study discovered and validated that methylation of *KCNQ1*, *KCNQ1OT1*, and *PHLDA2* at chromosomes 11p15.4-15.5 in peripheral blood leukocytes is associated with BC risk. The methylation of the *KCNQ1OT1* region could be applied as a potential biomarker for BC risk assessment.

## Figures and Tables

**Figure 1 cancers-14-02652-f001:**
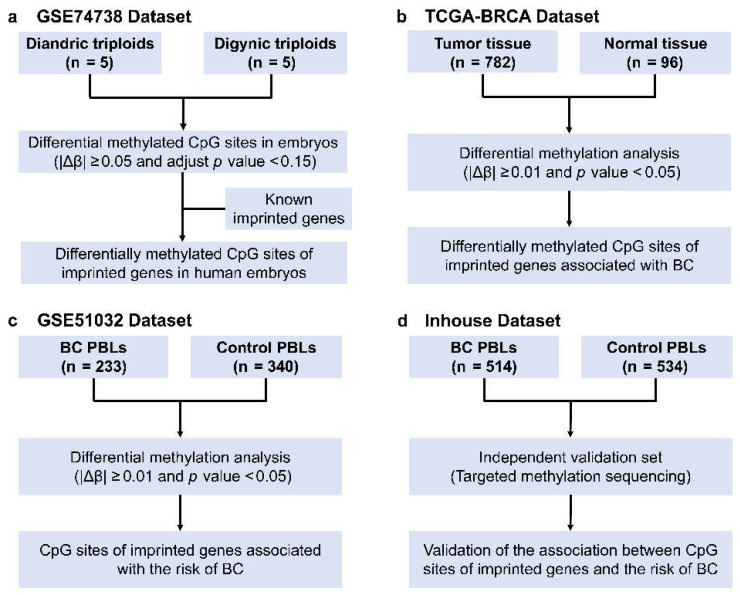
Overall workflow for identification and validation in different datasets. BC, breast cancer; PBLs, peripheral blood leukocytes, TCGA, The Cancer Genome Atlas. (**a**) Differential methylation analyses between diandric and digynic triploids. (**b**) Differential methylation analyses in selected CpG sites between 782 BC tissues and 96 normal tissues in the TCGA dataset. (**c**) Differential methylation analyses in selected CpG sites between 233 BC patients and 340 healthy controls in the GSE51032 dataset. (**d**) Validation of the association between CpG sites of imprinted genes and the risk of BC in an independent case-control study.

**Figure 2 cancers-14-02652-f002:**
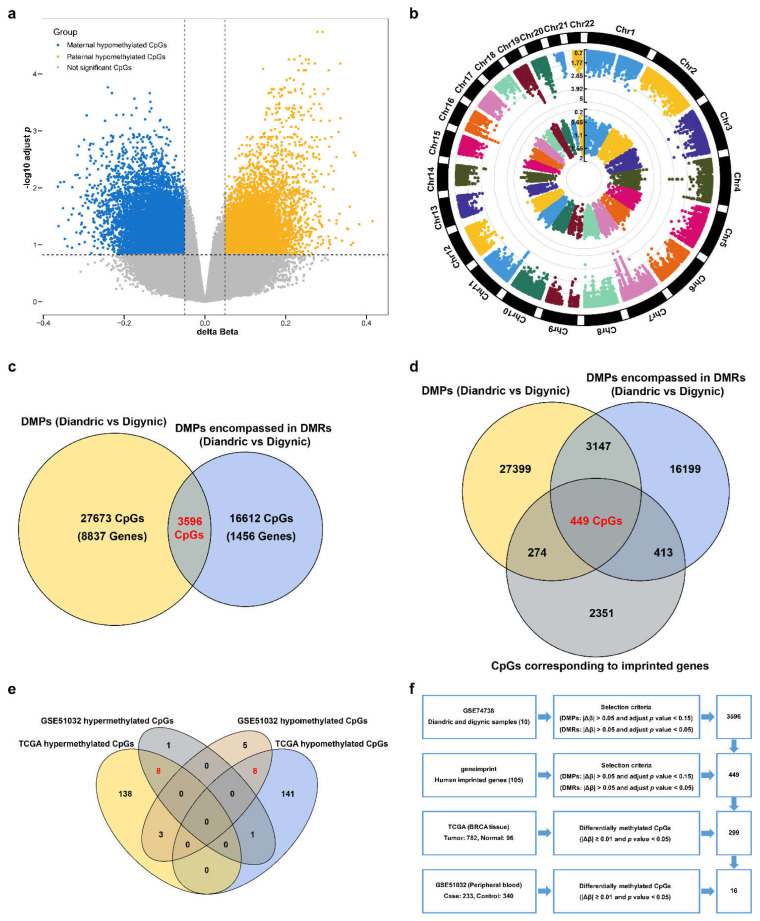
Discovery of the CpG sites of imprinted genes associated with the risk of breast cancer. DMP: differentially methylated position; DMR: differentially methylated region. (**a**) Differential methylation analysis comparing diandric and digynic triploids in the GSE74738 dataset. A volcano plot shows the -log10-adjusted *p*-value against the log2 delta beta for each CpG site. (**b**) Manhattan plot showing the chromosome (chr) and positional (pos) information for DMPs in the GSE74738 dataset. Dots in the inner ring represent -log10 |Δβ|, and dots in the outer ring represent -log10 FDR. (**c**) A two-way Venn diagram showing the intersection of DMPs from differential methylation analysis comparing diandric and digynic triploids and CpG sites encompassed in DMRs. (**d**) A three-way Venn diagram showing the intersection of DMPs, CpG sites encompassed in DMRs, and CpG sites corresponding to imprinted genes. (**e**) A four-way Venn diagram showing the intersection of differentially methylated CpG sites in the TCGA and GEO datasets (GSE51032). (**f**) A workflow for identification of the CpG sites of imprinted genes associated with the risk of breast cancer.

**Figure 3 cancers-14-02652-f003:**
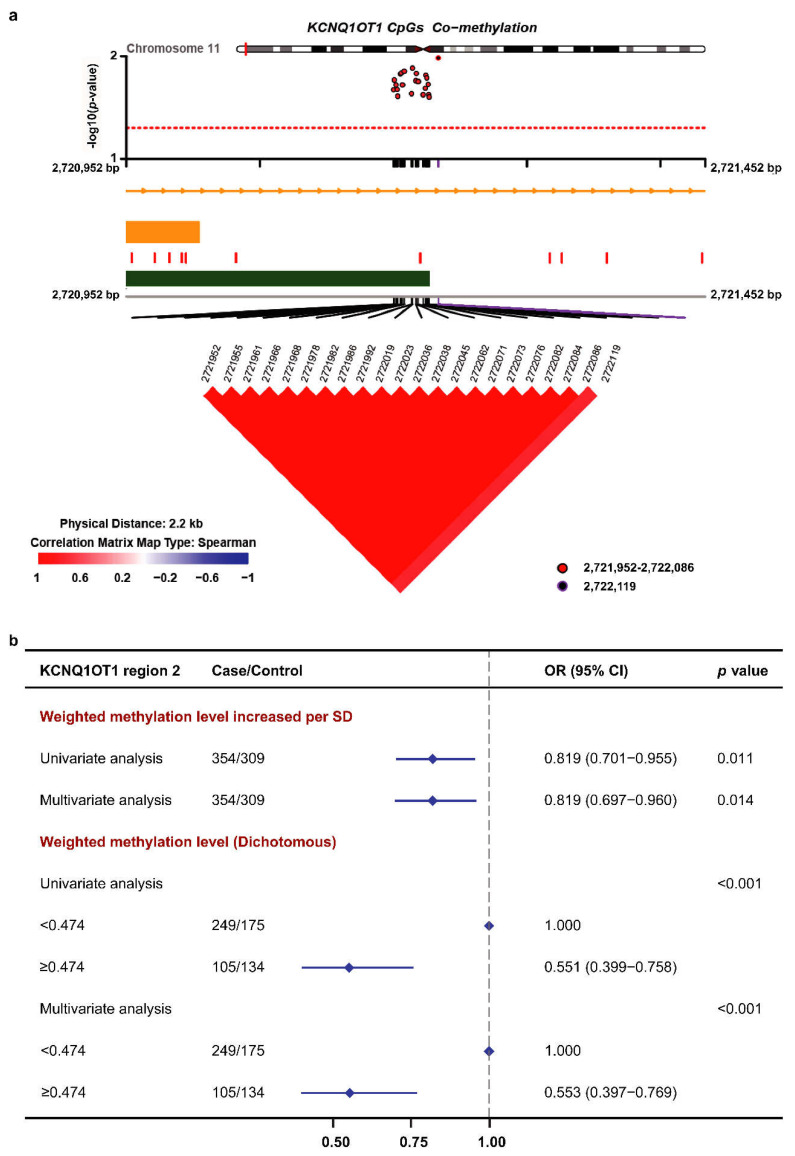
The association between *KCNQ1OT1* region methylation and the risk of breast cancer in the validation dataset. SD, standard deviation. (**a**) Co-methylation pattern of 22 CpG sites in *KCNQ1OT1* region 2. The co-methylation pattern was determined using Spearman correlation analysis. (**b**) The association between the weighted methylation level of *KCNQ1OT1* region 2 and the risk of breast cancer in the validation dataset. ORs of multivariate analyses were adjusted for age, BMI, race, location, family history of other cancers, and family history of breast cancer.

**Figure 4 cancers-14-02652-f004:**
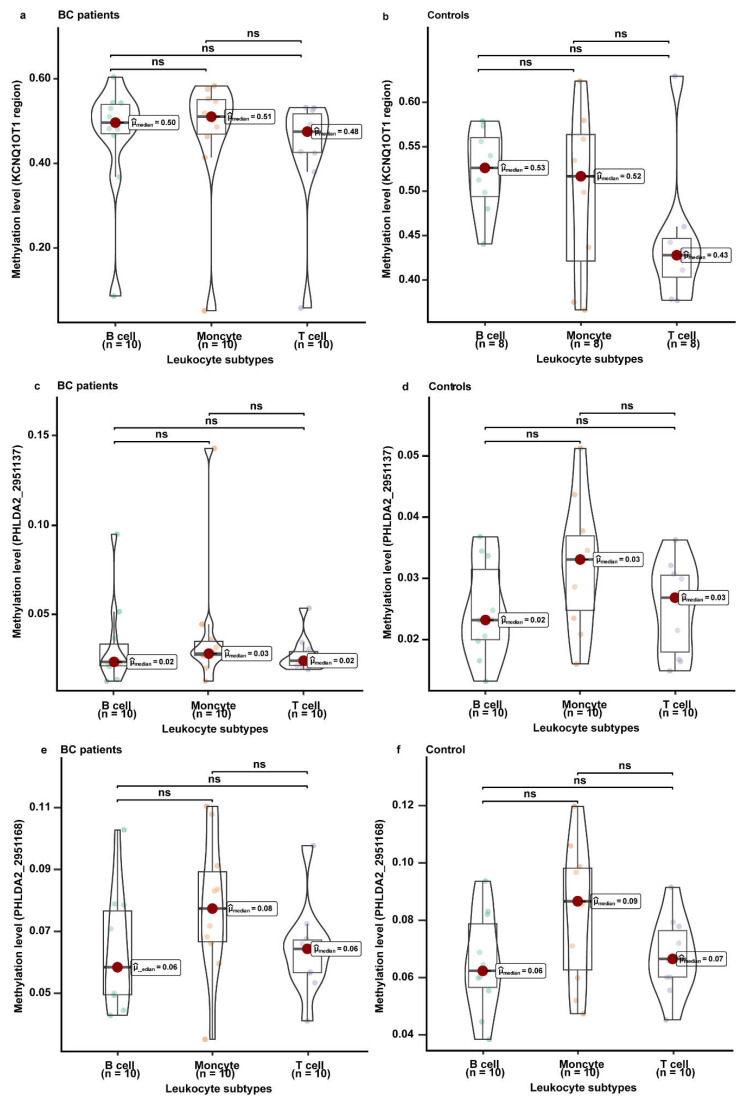
The methylation levels of the *KCNQ1OT1* region and *PHLDA2* in B cells, monocytes, and T cells. The methylation levels were measured in the samples (bisulfite-converted DNA from B cells, monocytes, and T cells) from 10 breast cancer patients and 10 healthy controls. The methylation levels between different leukocyte subtypes were compared using the *Kruskal–Wallis* test. (**a**) The methylation levels of the *KCNQ1OT1* region in B cells, monocytes and T cells from BC cases. (**b**) The methylation levels of the *KCNQ1OT1* region in B cells, monocytes and T cells from healthy controls. (**c**) The methylation levels of the CpG site with genome coordinates 2,951,137 located at *PHLDA2* in B cells, monocytes and T cells from BC cases. (**d**) The methylation levels of the CpG site with genome coordinates 2,951,137 located at *PHLDA2* in B cells, monocytes and T cells from healthy controls. (**e**) The methylation levels of the CpG site with genome coordinates 2,951,168 located at *PHLDA2* in B cells, monocytes and T cells from BC cases. (**f**) The methylation levels of the CpG site with genome coordinates 2,951,168 located at *PHLDA2* in B cells, monocytes and T cells from healthy controls.

**Table 1 cancers-14-02652-t001:** Demographic characteristics of breast cancer patients and controls.

Characteristics	Case (%)	Control (%)	*p*
Number of Participants	514	534	
Age (year) ^a,b^	52.0 ± 9.35	52.1 ± 10.76	0.908
<50	215 (41.8)	237 (44.4)	0.003
50–	192 (37.4)	147 (27.5)	
60–	89 (17.3)	122 (22.8)	
≥70	18 (3.5)	28 (5.3)	
BMI (kg/m^2^) ^a,b^	23.9 ± 3.65	23.7 ± 4.10	0.367
<18.5	15 (2.9)	27 (5.1)	0.325
18.5–	272 (52.9)	272 (50.9)	
24–	134 (26.1)	133 (24.9)	
≥28	93 (18.1)	102 (19.1)	
Ethnicity ^b^			0.139
Han	502 (97.7)	513 (96.1)	
Other	12 (2.3)	21 (3.9)	
Location ^b^			0.440
City	417 (81.1)	443 (83.0)	
Rural area	97 (18.9)	91 (17.0)	
Family history of other cancers ^b^			<0.001
No	392 (76.3)	454 (85.0)	
Yes	122 (23.7)	80 (15.0)	
Family history of breast cancer ^b^			<0.001
No	473 (92.0)	526 (98.5)	
Yes	41 (8.0)	8 (1.5)	

^a^ mean (standard deviation), ^b^ n (%).

**Table 2 cancers-14-02652-t002:** The association between the CpG site methylation of *KCNQ1OT1* and the risk of breast cancer in the validation dataset.

CpG_Position	Δβ	*p* (*t*-Test)	Univariate Analysis		Multivariate Analysis	
			OR (95% CI)	*p*	OR (95% CI) ^a^	*p*
*KCNQ1OT1*_2721952	−0.011	0.022	0.834 (0.714–0.973)	0.021	0.835 (0.711–0.978)	0.026
*KCNQ1OT1*_2721955	−0.012	0.017	0.829 (0.709–0.966)	0.017	0.828 (0.705–0.970)	0.020
*KCNQ1OT1*_2721961	−0.012	0.019	0.831 (0.712–0.969)	0.019	0.828 (0.705–0.971)	0.021
*KCNQ1OT1*_2721966	−0.012	0.021	0.834 (0.714–0.972)	0.021	0.834 (0.710–0.977)	0.025
*KCNQ1OT1*_2721968	−0.011	0.025	0.838 (0.717–0.977)	0.025	0.836 (0.712–0.979)	0.027
*KCNQ1OT1*_2721978	−0.012	0.015	0.825 (0.706–0.962)	0.015	0.823 (0.701–0.965)	0.017
*KCNQ1OT1*_2721982	−0.012	0.015	0.825 (0.706–0.962)	0.015	0.822 (0.700–0.963)	0.016
*KCNQ1OT1*_2721986	−0.012	0.019	0.831 (0.711–0.969)	0.019	0.828 (0.705–0.971)	0.021
*KCNQ1OT1*_2721992	−0.012	0.014	0.824 (0.705–0.961)	0.014	0.820 (0.698–0.961)	0.015
*KCNQ1OT1*_2722019	−0.011	0.023	0.836 (0.716–0.975)	0.023	0.836 (0.712–0.979)	0.027
*KCNQ1OT1*_2722023	−0.012	0.013	0.822 (0.704–0.959)	0.013	0.820 (0.698–0.961)	0.015
*KCNQ1OT1*_2722036	−0.012	0.017	0.829 (0.710–0.967)	0.017	0.829 (0.706–0.971)	0.021
*KCNQ1OT1*_2722038	−0.012	0.015	0.825 (0.706–0.962)	0.015	0.823 (0.701–0.964)	0.017
*KCNQ1OT1*_2722045	−0.012	0.018	0.829 (0.710–0.967)	0.018	0.828 (0.705–0.970)	0.020
*KCNQ1OT1*_2722062	−0.011	0.024	0.837 (0.717–0.976)	0.024	0.835 (0.711–0.979)	0.027
*KCNQ1OT1*_2722071	−0.011	0.020	0.833 (0.713–0.971)	0.020	0.832 (0.708–0.975)	0.023
*KCNQ1OT1*_2722073	−0.012	0.015	0.826 (0.707–0.963)	0.015	0.823 (0.701–0.965)	0.017
*KCNQ1OT1*_2722076	−0.012	0.016	0.828 (0.708–0.965)	0.016	0.824 (0.701–0.965)	0.017
*KCNQ1OT1*_2722082	−0.012	0.019	0.831 (0.711–0.969)	0.019	0.829 (0.705–0.971)	0.021
*KCNQ1OT1*_2722084	−0.011	0.024	0.837 (0.717–0.976)	0.024	0.834 (0.710–0.977)	0.025
*KCNQ1OT1*_2722086	−0.011	0.025	0.838 (0.718–0.977)	0.025	0.833 (0.709–0.976)	0.025
*KCNQ1OT1*_2722119	−0.031	0.009	0.814 (0.693–0.951)	0.010	0.834 (0.707–0.979)	0.028

Δβ represents the differential methylation of CpG sites between breast cancer patients and controls from the validation dataset, and the ORs were calculated based on the methylation levels increasing per standard deviation. ^a^ ORs of multivariate analyses were adjusted for age, BMI, race, location, family history of other cancers, and family history of breast cancer.

**Table 3 cancers-14-02652-t003:** The association between *KCNQ1OT1* region methylation and the risk of breast cancer in different subgroups.

Subgroup	*KCNQ1OT1* Region 2	Univariate Analysis		Multivariate Analysis	
		OR (95% CI)	*p*	OR (95% CI) ^a^	*p*
Age (year)					
<50	<0.474	1.000	<0.001	1.000	<0.001
	≥0.474	0.428 (0.265–0.687)		0.432 (0.263–0.703)	
≥50	<0.474	1.000	0.122	1.000	0.172
	≥0.474	0.701 (0.447–1.099)		0.728 (0.460–1.150)	
Molecular subtype					
Luminal A	<0.474	1.000	<0.001	1.000	<0.001
	≥0.474	0.449 (0.282–0.702)		0.446 (0.276–0.706)	
Luminal B	<0.474	1.000	0.085	1.000	0.107
	≥0.474	0.696 (0.458–1.048)		0.698 (0.450–1.080)	
HER-2	<0.474	1.000	0.061	1.000	0.042
	≥0.474	0.484 (0.216–1.004)		0.424 (0.176–0.936)	
Basal-like	<0.474	1.000	0.315	1.000	0.420
	≥0.474	0.653 (0.272–1.465)		0.705 (0.289–1.614)	

OR, odds ratio; 95% CI, 95% confidence interval. ^a^ ORs were calculated based on the methylation levels of CpG sites increased per standard deviation and were adjusted for age, BMI, race, location, family history of other cancers, and family history of breast cancer.

## Data Availability

The discovery datasets used to support the results of this study are available from The Cancer Genome Atlas (TCGA) database (https://xena.ucsc.edu/, accessed on 24 December 2021) and the Gene Expression Omnibus (GEO) database (https://www.ncbi.nlm.nih.gov/geo/, accessed on 30 December 2021). The validation case-control study generated during the current study is available from the corresponding author on reasonable request.

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
