# Peer review of "DNA Methylation of Imprinted Genes KCNQ1, KCNQ1OT1, and PHLDA2 in Peripheral Blood Is Associated with the Risk of Breast Cancer"

_cancers, 2022, doi:10.3390/cancers14112652_

Round 1
Reviewer 1 Report
In the proposed manuscript, titled “DNA methylation of imprinted genes KCNQ1, KCNQ1OT1, and PHLDA2 in peripheral blood is associated with the risk of breast cancer”, the authors reported an experimental study aimed to first identify the CpG sites of imprinted genes, associated with breast cancer (BC) risk. They also described an independent case-control study to validate DNA methylation of these CpG sites of KCNQ1, KCNQ1OT1, and PHLDA2 genes, in peripheral blood and BC susceptibility. The authors conclude that the methylation of KCNQ1, KCNQ1OT1, and PHLDA2 is associated with BC risk. Moreover, KCNQ1OT1 methylation is an attractive potential biomarker for BC risk assessment. The manuscript might be interesting and relevant to the cancer research field and medical research community; however, it requires any minor revisions and changes before publication. There are some relevant formal and technical issues that must be addressed.
- At first, the introduction of the paper is extremely poor, and it does not provide sufficient information to the reader about the topic of the work. The authors should totally rewrite the introduction.
- The authors should make a general revision of the manuscript in order to make the manuscript easier to read. For instance, some general information in the “Discussion” section, should be reported in the introduction and so on…
- The quality of the figures reported by the authors is not so good. For instance, in Figures 3 and 4, the axis titles, and the size of data in the graphs are too small! It is very difficult for the reader to understand. The authors should find a better way to report all the figures/graphs of the paper.
- I really appreciated the innovative approach in the search for risk factors associated with BC. However, I believe that some of the authors' statements, such as “the methylation of KCNQ1, KCNQ1OT1, and PHLDA2 was associated with BC risk. KCNQ1OT1 methylation is an attractive potential biomarker for BC risk assessment.”(lines 42-43) too risky. I believe that the data presented in the work are promising but should be confirmed by further and future studies. The authors should mitigate their claims.
Reviewer 2 Report
Thank you for your valuable work, yet there are some flaws to be addressed:
- The manuscript should be revised by a native English speaker
- Please improve the quality of figures 3 and 4, that are too small.
- Please clarify the clinical impact of your findings
Author Response
Response to Reviewer 2 Comments
Point 1: The manuscript should be revised by a native English speaker.
Response 1: Thank you for your kind and valuable advice. We followed your advice and have finished the manuscript revision by using the language editing services from AJE Corporation. Your advice improved the clarity of this manuscript and make it easier to read.
Point 2: Please improve the quality of figures 3 and 4, that are too small.
Response 2: Thank you for your insightful suggestion. We feel sorry that Figures 3 and Figures 4 are too small to read and understand. We have revised these figures and improved their quality. In order to display the results more clearly, we also revised parts a and b in Figure 3 into Table 2.
Point 3: Please clarify the clinical impact of your findings.
Response 3: Thank you for your kind advice. We followed your advice and added a paragraph in the Discussion to describe the clinical impact of our findings (see line 413-425 on page 15).

Reviewer 3 Report
The authors investigated the CpG sites of imprinted genes that could be associated with the risk of breast cancer and found that the methylation level of KCNQ1, KCNQ1OT1, and PHLDA2 could be associated with breast cancer risk
The criteria used for detecting differentially methylated sites were arbitrarily selected. The cutoff parameter |Δβ| that the authors chose was not a good enough for methylation state comparison since the difference was too small. Also, the plausibility of performing t-test and Pearson correlation on methylation beta values should be stated since the violation of normality assumption can often be observed in methylation data.
- Line 204. The authors stated that 31,269 differentially methylated CpG sites were found. But Hanna et al, the contributor of the dataSets GSE74738, only found 6,807 differentially methylated sites (PMID: 26769960), the difference between these two results should be clearly stated.
- Figure2a Y-axis title should be -log10 adjust P, line 232, log10 FDR or - log10 FDR?
- Line 278-280. I’m wondering why the authors stated that the methylation level of cg25306939 was significantly lower in BC patients. It seems no difference for me if the Δβ=-0.011. Also, Δβ ranging from -0.031 to -0.011 would not mean the difference, either.
- Line 288, I don't agree with the authors’ conclusion as I mentioned above.
- Line 299-300, “weighted methylation level 299 lower than 0.474 (hypomethylated group) “, why the authors chose 0.474 should be clearly stated since the result from line 300 to line 331 was largely based on it.
- Figure 4 what does Y-axis refer to
- Line 406, the authors should investigate the KCNQ1OT1 gene expression level using TCGA dataset before drawing a conclusion
- Line 437 “systematically screened the BC-associated imprinted genes based on high-throughput sequencing data”. the authors used methylation array but not high-throughput sequencing data.
Author Response
Response to Reviewer 3 Comments
Point 1: The criteria used for detecting differentially methylated sites were arbitrarily selected. The cutoff parameter |Δβ| that the authors chose was not a good enough for methylation state comparison since the difference was too small. Also, the plausibility of performing t-test and Pearson correlation on methylation beta values should be stated since the violation of normality assumption can often be observed in methylation data.
Response 1: Thank you for your careful work and valuable question. We reanalyze the data from TCGA and GEO datasets using t-test with the criterion, that is P value less than 0.05 and the absolute value of the average methylation difference between cases and controls (|Δβ|) greater than 0.01.
Normally, differential methylation analysis of each CpG site was tested by moderated t statistics. The differentially methylated CpG sites were defined as those having FDR with the Benjamini-Hochberg procedure of less than 0.05. However, in our research, the differences in methylation levels of CpG sites of imprinted genes between the BC patients and the controls in the GEO dataset were relatively small. When we conducted the above method using “ChAMP” or “minfi” package, the number of screened differentially methylated CpG sites is very small.
A systematic review revealed that the overall methylation difference in blood DNA between BC cases and controls is relatively small (effect size varied from 0.013 to 0.25) [1]. Moreover, Severi G et al. carried out a nest case-control study using Illumina Infinium Human Methylation 450 BeadChip array from peripheral blood to detect the biomarker for BC risk. The results showed that epigenome-wide methylation was lower for cases than controls and the difference between cases and controls was small (difference between cases and controls: -0.0011; P=0.006) [2]. The studies revealed a small difference in methylation between BC cases and controls in blood.
Based on the characterization of the methylation in peripheral blood, we directly analyzed the data from GEO datasets using t-test and calculated the absolute value of the average methylation difference between cases and controls (|Δβ|). In our analysis, |Δβ| values ranged from 0.002 to 0.023 based on the GEO datasets, and the 95th percentile of |Δβ| was close to 0.010. Differentially methylated CpG sites with |Δβ| values in the top 5% were selected for the following study. Therefore, we defined the threshold as |Δβ| value <0.01 and P value<0.05.
Actually, when performing t-test and correlation analysis, it is required that the data conform to a normal distribution. In our study, TCGA (878 samples), GEO (573 samples), and inhouse dataset (1048 samples) contained large sample sizes. Therefore, for data containing large samples, we can approximate it as normally distributed data. We hope that the explanation will meet with approval.
Point 2: Line 204. The authors stated that 31,269 differentially methylated CpG sites were found. But Hanna et al, the contributor of the dataSets GSE74738, only found 6,807 differentially methylated sites (PMID: 26769960), the difference between these two results should be clearly stated.
Response 2: Thank you for your careful work and valuable question. In our study, differentially methylated CpG site was defined as a CpG with the absolute value of methylation difference (|Δβ|) ≥0.05 and adjusted P-value <0.15. Based on the above selection criteria, 31269 differentially methylated CpG sites were found. However, in the study of Hanna et al (the contributor of the dataSets GSE74738), 6,807 differentially methylated sites were encompassed in 882 differentially methylated regions which were defined as containing at least three differentially methylated CpG sites within 500bp. While those differentially methylated CpG sites that were not encompassed in the differentially methylated regions were excluded in their study. Therefore, I think the difference between these two results can be explained by the different selection criteria of differentially methylated CpG sites in these two studies. We hope that the explanation will meet with approval.
Point 3: Figure2a Y-axis title should be -log10 adjust P, line 232, log10 FDR or - log10 FDR?
Response 3: We are very grateful for your excellent suggestions. We feel sorry that we made a silly mistake. We have revised Figure 2 and corrected the Figure2a Y-axis title to -log10 adjust P. We also corrected the log10 FDR for line 232 to -log10 FDR (see line 239 on page 8).
Point 4: Line 278-280. I’m wondering why the authors stated that the methylation level of cg25306939 was significantly lower in BC patients. It seems no difference for me if the Δβ=-0.011. Also, Δβ ranging from -0.031 to -0.011 would not mean the difference, either. Line 288, I don't agree with the authors’ conclusion as I mentioned above.
Response 4: Thank you for your comment. Several studies have revealed that the methylation differences in peripheral blood between BC cases and controls are small. A systematic review revealed that the overall methylation difference in blood DNA between BC cases and controls are relatively small (effect size varied from 0.013 to 0.25) [1]. Moreover, Severi G et al. carried out a nest case-control study using Illumina Infinium Human Methylation 450 BeadChip array from peripheral blood to detect the biomarker for BC risk. The results showed that epigenome-wide methylation was lower for cases than controls and the difference between cases and controls was small (difference between cases and controls: -0.0011; P=0.006) [2]. In the validation dataset of our study, the methylation level of cg25306939 located at KCNQ1OT1 was statistically lower in BC patients than that in controls (Δβ=-0.011). We agree with you that although the methylation level of single CpG sites is statistically different, the difference is not significant. our study found that although the methylation differences of KCNQ1OT1 region between cases and controls were small, it also had the ability to predict breast cancer susceptibility. In further study, we will investigate the mechanism by which methylation differences in peripheral blood play a role in breast cancer development. We revised the manuscript, removing the word “significant”, to make the description more accurate. In addition, we also complement the limitation that the differences in methylation were relatively small in our study (see line 297-300 on page 9). We hope that the explanation will meet with approval.
Point 5: Line 299-300, “weighted methylation level 299 lower than 0.474 (hypomethylated group) “, why the authors chose 0.474 should be clearly stated since the result from line 300 to line 331 was largely based on it.
Response 5: Thank you for your kind advice. In our study, the optimal cut-off values of methylation level were calculated using the receiver operating characteristic curve (see line 191-194 on page 6). Therefore, 0.474 as the optimal cut-off value for weighted methylation of KCNQ1OT1 region is also based on the receiver operating characteristic curve (see Table S11). To make the results more clear, we added Table S11 and a short description to explain why “subjects were separated into the hypomethylated group (<0.474) and hypermethylated group (≥0.474)” (see line 327-332 on page 11).
Point 6: Figure 4 what does Y-axis refer to.
Response 6: Thank you for your kind question. In our study, Figure 4 Y-axis refer to the methylation level of targeted region and CpG sites in different leukocyte subtypes. We feel sorry that Figure 4 The Y-axis is too vague, making it very difficult for the reader to understand. We have revised Figure 4 and corrected Figure 4 Y-axis. Thank you again.
Point 7: Line 406, the authors should investigate the KCNQ1OT1 gene expression level using TCGA dataset before drawing a conclusion.
Response 7: Thank you for your valuable advice. In our study, the results of differential methylation analyses using TCGA, GEO, and inhouse datasets showed that the methylation level of cg25306939 located at KCNQ1OT1 was lower in BC tissue than that in normal adjacent breast tissue; lower cg25306939 methylation was also observed in peripheral blood of BC patients. These results have been reported in Table 2 and Table S3. To explore the potential regulatory function of selected CpG sites, the Pearson correlation method was also applied to assess the correlation between the methylation of cg25306939 located at KCNQ1OT1 and its corresponding expression in the TCGA dataset. The results showed that the methylation level of cg25306939 was negatively correlated with its gene expression, which could be found in Figure S2E. Overall, lower cg25306939 methylation and higher KCNQ1OT1 expression existed in BC patients, which implied hypomethylation of KCNQ1OT1 region (located in the promoter) may results in its high expression, thereby contributing to BC development. Therefore, our study suggested that KCNQ1OT1 methylation could be a novel, potential biomarker for predicting BC risk.
Point 8: Line 437 “systematically screened the BC-associated imprinted genes based on high-throughput sequencing data”. the authors used methylation array but not high-throughput sequencing data..
Response 8: Thank you for your kind advice. We are sorry that the description in our study was not accurate enough. We followed your advice and have revised sequencing data to array data (see line 481-483 on page 16).
- Tang, Q.; Cheng, J.; Cao, X.; Surowy, H.; Burwinkel, B. Blood-based DNA methylation as biomarker for breast cancer: a systematic review. Clinical epigenetics 2016, 8, 115, doi:10.1186/s13148-016-0282-6.
- Severi, G.; Southey, M.C.; English, D.R.; Jung, C.H.; Lonie, A.; McLean, C.; Tsimiklis, H.; Hopper, J.L.; Giles, G.G.; Baglietto, L. Epigenome-wide methylation in DNA from peripheral blood as a marker of risk for breast cancer. Breast Cancer Res Treat 2014, 148, 665-673, doi:10.1007/s10549-014-3209-y.

Reviewer 4 Report
Comments:
- Are KCNQ1, KCNQ1OT1, and PHLDA2 paternal or maternal imprinting genes?
- The introduction needs to add more info on KCNQ1, KCNQ1OT1, and PHLDA2 information.
- Explain why BMI cutoff is > 27?
- What is the definition of rural area in the current study?
- What is the difference between city and rural area?
- On Table 1, family history of other cancers? What are other cancers? Any specific data on ovarian and endometrial cancers?
Author Response
Response to Reviewer 4 Comments
Point 1: Are KCNQ1, KCNQ1OT1, and PHLDA2 paternal or maternal imprinting genes?
Response 1: Thank you for your insightful question. KCNQ1, KCNQ1OT1, and PHLDA2 are known imprinted genes. KCNQ1 and PHLDA2 are paternal imprinted genes, as their maternal alleles were expressed but paternal alleles were not expressed. While KCNQ1OT1 is maternal imprinted gene, its paternal alleles were expressed but maternal alleles were not expressed.
Point 2: The introduction needs to add more info on KCNQ1, KCNQ1OT1, and PHLDA2 information.
Response 2: Thank you for your suggestion. We feel very sorry that we ignored describe the information on KCNQ1, KCNQ1OT1, and PHLDA2 in the Introduction section. Your suggestion gives us a valuable reminder, and we have added a sufficient description of KCNQ1, KCNQ1OT1, and PHLDA2 in the Introduction (see line 51-92 on page 3).
Point 3: Explain why BMI cutoff is > 27?
Response 3: Thank you very much for your comments. As all the subjects come from China, We used the Chinese BMI evaluation criteria to divide them into four groups: underweight (BMI <18.5), normal weight (18.5≤ BMI <24), overweight (24≤ BMI <28), and obese (BMI ≥28). We apologize for the mistake of writing the cutoff value for BMI as 27. We have carefully checked the tables in the manuscript and supplementary files, and have corrected the BMI cutoff from 27 to 28. Your detailed comments helped us to improve the quality of this manuscript. Thank you.
Point 4: What is the definition of rural area in the current study?
Response 4: Thank you for your comment. Study shows that environmental factors can affect the methylation levels of specific genes and whole genes in peripheral blood leukocytes [1, 2]. Considering the differences in environmental factors such as fertility, diet, and lifestyle between urban and rural women in China, the proportion of urban and rural populations in the case and control groups may affect the analysis of imprinted gene methylation and breast cancer susceptibility. Therefore, in this study, we obtained the participants' detailed resident addresses through a questionnaire by trained investigators. The city and rural areas were divided based on participants' resident addresses. Those participants living in urban areas above the town and county level are classified as urban residents, and the rest are classified as rural residents.
Point 5: What is the difference between city and rural area?
Response 5: Thank you for your comment. In China, women in urban and rural areas were different in many aspects, such as education, age at first birth, number of deliveries, eating habits and lifestyle, etc. All these environmental factors can affect the methylation levels of specific genes in peripheral blood. Our study divided participants into urban and rural areas based on their resident addresses and compared the proportion of urban and rural populations in the case and control groups. The results showed no significant difference in the proportion of urban and rural populations between the case and control groups, which also implied a better matching between case and control groups. We sincerely hope that the explanation will meet with approval.
Point 6: On Table 1, family history of other cancers? What are other cancers? Any specific data on ovarian and endometrial cancers?
Response 6: Thank you for your valuable comment. Compared with general population, people with a family history of breast cancer have a higher incidence of breast cancer. In addition to a family history of breast cancer, a family history of other cancers may also increase the risk of breast cancer. Therefore, our study collected both the family history of breast cancer and other cancers (including stomach cancer, esophageal cancer, liver cancer, colorectal cancer, ovarian cancer, endometrial cancer, and other cancers) from all the participants. Compared with controls, BC patients had a higher proportion of family history of BC and other cancers. In subsequent multivariate logistic regression, family history of breast cancer and family history of other cancers were adjusted.
References:
- Chiarella J, Tremblay R E, Szyf M, Provençal N, Booij L. Impact of Early Environment on Children's Mental Health: Lessons From DNA Methylation Studies With Monozygotic Twins. Twin Res Hum Genet, 2015, 18(6):623-634.
- Yet I, Tsai P C, Castillo-Fernandez J E, Carnero-Montoro E, Bell J T. Genetic and environmental impacts on DNA methylation levels in twins. Epigenomics, 2016, 8(1):105-117.

Round 2
Reviewer 1 Report
Dear Authors,
thank you for your answer. I really appreciated the changes you made to the paper, especially in the introduction. I think that the proposed paper can be published on Cancer.
Best regards
Reviewer 2 Report
The authors addressed all the comments, therefore I consider the paper suitable for publication.
Reviewer 3 Report
I have no further comments.
Reviewer 4 Report
No more comments.